# The Therapeutic Potential of West Indian Lemongrass (*Cymbopogon citratus*) Essential Oil-Based Ointment in the Treatment of Pitted Keratolysis

**DOI:** 10.3390/antibiotics14030241

**Published:** 2025-02-27

**Authors:** György Schneider, Bettina Schweitzer, Anita S. Steinbach, Ágnes S. Hodován, Marianna Horváth, Eszter Bakó, Anna Mayer, Szilárd Pál

**Affiliations:** 1Department of Medical Microbiology and Immunology, Medical School, University of Pécs, Szigeti st. 12., H-7624 Pécs, Hungary; sbetti101@gmail.com (B.S.); seres-steinbach.anita@edu.pte.hu (A.S.S.); agi.hodovan@pte.hu (Á.S.H.); horvath.marianna2@pte.hu (M.H.); 2Department of Pharmacognosy, Semmelweis University, Üllői st. 26., H-1085 Budapest, Hungary; bakoeszterdr@gmail.com; 3Centre for Translational Medicine, Semmelweis University, Üllői st. 26., H-1085 Budapest, Hungary; 4Institute of Clinical Pharmacy, Medical School, University of Pécs, Szigeti st. 12., H-7624 Pécs, Hungary; mayer.anna@pte.hu; 5Institute of Pharmaceutical Technology and Biopharmacy, Faculty of Pharmacy, University of Pécs, Rókus str. 2., H-7624 Pécs, Hungary; pal.szilard@gytk.pte.hu

**Keywords:** Indian lemongrass, pitted keratolysis, essential oil-based ointment

## Abstract

**Background:** Due to their antibacterial activities, essential oils can be potential alternatives to antibiotics in certain cases. West Indian lemongrass (*Cymbopogon citratus*) essential oil (LEO) is effective against a broad range of bacteria by inhibiting spore formation, and is considered safe. In this study, we demonstrated its therapeutical potential in the treatment of pitted keratolysis (PK), a superficial skin infection affecting the pressure-bearing areas of plantar surfaces. **Methods:** For in vitro antibacterial efficacy testing, LEO was mixed into different ointment bases, including Hydrogelum methylcellulose FoNo VIII., Ungentum oleosum FoNo VIII. (Ung. oleoso), Unguentum stearini FoNo VIII. (Ung. stearin), and Vaselinum cholesterinatum FoNo VIII. (Vasel. cholest.), at different concentrations of 1, 3, and 5%. These formulations were tested on representatives of three bacterial species associated with PK: *Kytococcus sedentarius*, *Dermatophilus congolensis*, and *Bacillus thuringiensis*. **Results:** In the in vitro tests, Hydrogelum methylcellulose (HM) gel best supported the antibacterial effects of LEO, reducing the number of living bacteria on agar plates by 4–5 orders of magnitude in a concentration-dependent manner during the 30 min exposure times. This was also confirmed by the Franz diffusion cell drug release test; after 30 min, several active compounds could be detected in the HM samples, in contrast to the other bases. Shelf-life experiments showed that the HM base supported the antibacterial features of 3% LEO for at least 2 years without significant loss of efficacy. **Conclusions:** Our study highlights that ointments containing essential oils potentially have a place in the treatment of PK. Therefore, antibiotics may potentially be replaced for the treatment of PK, thereby reducing environmental antibiotic pressure, which is one of the driving forces behind the spread of antibiotic resistance.

## 1. Introduction

The spread of antimicrobial resistance (AMR) is one of the greatest global health challenges of our time. It is estimated that 1.3 million deaths worldwide were associated with AMR in 2019, but this number is expected to rise to more than 10 million by 2050 [1]. The spread of AMR and the consequent emergence of new antimicrobial-resistant microorganisms are fuelled by several factors, the most important of which is the overuse of antibiotics in human and animal medicine and in crop production. As a result, resistant bacteria emerge and spread through wastewater, natural waters, and soil, and these sites have now become reservoirs for antibiotic-resistant bacteria and AMR genes [2,3].

To address this dilemma, it is necessary to reduce the use of antibiotics and slow down the accelerated process of AMR spread. A combined approach is urgently needed, requiring a change in attitudes and the widespread introduction of alternative treatments wherever possible [4]. Solutions could include the use of antiviral strategies [5], antibody therapies, antimicrobial peptides and nanoparticle-based therapeutics, phage therapy, and the utilization of the antibacterial features of phytochemicals [6].

Essential oils (EOs) are a group with potential as therapeutic agents. Numerous recent studies have investigated the biological and, by extension, antimicrobial effects of EOs [7]. Due to their relative ease of access and application, they are now considered potential therapeutic agents in modern medicine. West Indian lemongrass (*C. citratus*) contains an infinite number of active compounds and has clear antibacterial effects [8,9,10].

Skin surface infections are ideal targets for the introduction of EO-based therapeutics, as they are easily accessible by small and mostly hydrophobic compounds. Pitted keratolysis (PK) is a superficial bacterial skin infection that typically affects the pressure-bearing areas of the soles and rarely the palms [11]. Clinically, it is characterized by multifocal, discrete, superficial crateriform pits and erosions confined to the stratum corneum, as well as malodor. Wearing occlusive shoes and living in humid areas are the main predisposing factors, which is why athletes, sailors, soldiers, and citizens of flooded and tropical countries are the most commonly affected. Recently, a number of cases have also been reported in citizens of high socioeconomic status living in cities [12]. Hyperhidrosis, increased skin pH, poor foot hygiene, obesity, and certain immunodeficiencies are predisposing factors for PK [13]. *Kytococcus* (formerly *Micrococcus*) *sedentarius*, *Corynebacterium species*, *Dermatophilus congolensis*, *Actinomyces keratolytica*, *Streptomyces* spp., and *Bacillus thuringiensis* have been described as etiological agents of this infection [14]. In addition to their colonization potential, the protease production of these bacteria is a key factor contributing to the typical unpleasant symptoms of the infection. Spores may play a role in disease recurrence, as indicated by the identification of *Bacillus* species as potential etiological agents of PK.

The treatment of PK today is based on topical therapeutics, such as erythromycin-, tetracycline-, and clindamycin-containing ointments or oral doses of clindamycin and erythromycin in stubborn cases [12,15]. Antiseptics, such as glutaraldehyde and formaldehyde, can also be used, but they are not preferred, as they are toxic to the host eukaryotic cells themselves. Recently, a number of resistant cases and post-therapeutic relapses after topical treatments have been reported [12], highlighting the need for alternative therapies. 

In a recent study, we demonstrated the antibacterial effects of lemongrass essential oil (LEO) on the known etiological agents of PK [8]. Here, the concentration-dependent antibacterial efficacy of LEO was compared in base ointments with different hydrophobicities. An artificial skin model was used to determine the compound release capacities of the different ointment bases. Based on our findings, we evaluated the efficacy of the Hydrogelum methylcellulose ointment base containing 3% LEO in the potential treatment of PK in vivo.

## 2. Results

### 2.1. Gas Chromatography Analysis

Gas chromatographic analysis was used to determine the compound composition of West Indian lemongrass essential oil (LEO; Table 1). Seventeen different compounds were identified, with Neral (26.45%) and Geranial (27.06%) comprising the highest proportion. Other major components included p-cymen-ol (4.82%), Geranyl acetate (5.06%), Borneol (5.35%), Eucalyptol (5.48%), and α-terpineol 5.61%).

### 2.2. Antibacterial Testing of the LEO Containing Ointments

The antibacterial effects of increasing concentrations of LEO mixed into various ointment bases were determined on the surface of agar plates using a total of 11 different *Bacillus thuringiensis* isolates. Three parallel measurements were carried out in order to test the antibacterial efficacy of the different ointment bases, containing 1%, 3% and 5% LEO concentrations. The post hoc analysis was a comparison of bacteria, ointment type, and LEO concentration using Tukey’s test to assess the significance. CFUs were considered the dependent variable, while the fixed factors were bacteria, the concentration of LEO and the type of ointment. A value of *p* < 0.05 was considered significant. *p*-value was <0.01 by the following criteria: type of ointments, bacterium CFU and concentrations, as well as the type of ointments vs. bacteria, type of ointments vs. concentration, bacteria vs. concentration, and type of ointments vs. bacteria vs. concentration. 

The results showed that there was no significant reduction in colony forming units (CFUs) in samples where the ointment bases Vaselinum cholesterinatum and Ungentum oleosum were used as carriers (Figure 1). Ungentum stearini containing 3% or 5 % of LEO was effective against eight isolates (5%), with Hydrogelum methylcellulose showing the strongest reduction in CFU (add percentage). Detailed results of the pairwise comparisons can be found in Appendix A.

### 2.3. Shelf Life of LEO Containing Ointment

Shelf-life tests were carried out with Hydrogelum methylcellulose, based on its significant reduction in CFUs in vitro. Effect of the ointment on CFU of 13 bacteria after storage for 6–24 months at room temperature was assessed. Although a slight decrease in the antibacterial efficacy was detected at 1 %, 3 % and 5% LEO, the changes were not significantly significant (Figure 2).

### 2.4. Franz Diffusion Cell Drug Release and Solid Phase Microextraction–Gas Chromatography–Mass Spectrometry

The Franz diffusion cell drug release test was carried out to determine how four ointment bases retained or released the compounds of LEO, as detected by solid-phase microextraction–gas chromatography–mass spectrometry. Ointment mixtures containing 1 % LEO were used and the release of the solubilized compounds was recorded at multiple timepoints following incubation for up to 300 min. In the case of Vaselinum cholesterinatum, no identifiable compounds were detected in any of the samples (Table 2). In the case of Ungentum oleosum, no identifiable compounds could be detected at the 30th, 60th, 90th and 120th min sampling times, but by the 300th min of the circulation, β-caryophyllene was detected in the wash solution. Similarly, following washing of Ungentum stearini for 300 min, five compounds were detected: linalool (7%), derivatives of linalool (3–7%), camphor (1%) and beta-caryophyllene (4–9%). Hydrogelum had the lowest LEO retention capacity (Table 2) as linalool-oxide (2–5%), p-cymen-ol (12–18%) and α-terpineol (8–16%) could already be detected after 30 min. Additional compounds were detected at later time points (Figure 3; Table 2).

### 2.5. In Vivo Therapeutic Testing of Hydrogel Methylcellulose Ointment Containing 3% LEO

An in vivo test was conducted to assess the therapeutic efficacy of ointment containing LEO in the treatment of PK. The subject was a 43-year-old man with a long history of PK. Odor, sensitivity, itching and burning sensation were the main discomfort causing symptoms on the soles of the patient’s feet, especially under pressure.

Following one day of treatment, the intensity of malodor was drastically reduced, and on the third day, it had completely resolved (Table 3). All symptoms were resolved following 3 days of treatment. Microbiological examination on the fourth day of treatment revealed no presenting bacteria in the pits. Thirty days after the beginning of the treatment, none of the unpleasant symptoms had reappeared. No new pits were observed, and the smaller pits began to disappear, while the remnants of the deep confluent erosions took months to disappear (Figure 3).

## 3. Discussion

The use of alternative methods to antibiotic treatments is an epidemiological necessity, as the spread of antimicrobial resistance (AMR) is one of the most pressing global public health issues [1]. AMR may be combatted by partially replacing classical antibiotic treatments, thereby reducing their environmental presence. 

Due to its superficial presentation, PK is an ideal target for EO-based therapeutics. Although it is not a serious infection, patients report that symptoms become increasingly uncomfortable as the infection progresses and small pits coalesce into crateriform erosions (Figure 4). Recent reports of treatment failures and the relatively high prevalence of this sometimes unrecognized infection in multiple segments of society indicate that alternative therapies should be explored for the disease [12]. 

In this study, we assessed multiple EOs for the treatment of PK. LEO was the focus of our experiments because it has broad bioactivity, distinct antibacterial properties [8,10,16], can inhibit spore formation [8,17] and is generally recognized as safe (GRAS) [18,19]. We used gas chromatography to assess the chemical composition of LEO, which confirmed the presence of major and minor compounds in a similar ratio as previously published [20,21]. The chemical composition of EO batches must be evaluated, as antibacterial activities can be influenced by multiple environmental factors [22]. Confirmation of its antibacterial activity on *Dermatophilus congolensis, Kytococcus sedentarius*, and 11 natural isolates of *Bacillus thuringiensis* with extended protease activities, all of which are associated with PK [14] (Figure 1), strongly suggests that LEO is a good candidate for the treatment of PK and could contribute to the development of novel antibiotic strategies based on alternative therapeutics.

When considering the therapeutic potential of a candidate compound and compound mixture, the carrier medium or ointment base used is a crucial aspect. The four ointments (Table 4) involved in our study are basic ointments commonly used in dermatological practice. Their application is dependent on skin type and on the aim of the therapy. By incorporating different active compounds, they are used to treat multiple conditions, including hemorrhoids (Ungentum oleosum with lidocaine, Cliochinolon, and the EO *Matricaria chamomillae*), eczema keratolycum (Vaselinum cholesterinatum with ureum), and aphtae (Hydrogelum methylcellulose with camphor, menthol and capsici tintura or with metronidazole and lidocain). Crucially, the differing characteristics of the four basic ointments (Table 4) could influence the release of the antimicrobial active components of LEO (Table 2) and their concentration-dependent antibacterial activity (Figure 1). The ineffectiveness of Ungentum oleosum FoNo VIII. (Ung. oleoso), Unguentum stearini FoNo VIII. (Ung. stearin) and Vaselinum cholesterinatum FoNo VIII. (Vasel. cholest.) in the antibacterial tests (Figure 1) suggested that these ointment bases effectively retain the antibacterial active hydrophobic compounds of LEO in their matrix for up to 300 min, when the first released compounds were detected with GC-MS (Table 2). Thus, its antibacterial effect, previously demonstrated with the drop plate method [8] and here, could not be exerted.

In contrast, the pronounced antibacterial effect of LEO suspended in Hydrogelum methylcellulose (Figure 1) may be associated with the hydrophilic nature of the backbone molecule, which may have a minimal retention effect towards the hydrophobic compounds of LEO. Nevertheless, a mild retention effect could be attributed to the hydrophobic methyl groups, which slightly bind the hydrophobic compounds of LEO, mostly consisting of terpenes and terpenoids [19]. However, this retaining effect was drastically different from the retention effects of Ungentum oleosum, Unguentum stearini and Vaselinum cholesterinatum, as demonstrated in the Franz diffusion cell drug release model (Table 2), since the dissolution of the first compounds from the Hydrogelum methylcellulose could already be detected after 30 min of sampling. These compounds, including linalooloxide, p-cymenol and α-terpineol (Table 2), have recently been reported to have antibacterial activities [8,23,24]. Interestingly, in the case of the oil/water emulsion Ungentum stearini, retention was strong and comparable with Ungentum oleosum and Vaselinum cholesterinatum (Table 2), but it showed a significant antibacterial effect when applied to LB agar plates (Figure 1). One explanation for this apparent contradiction could be that in vitro, bacteria and the LEO-containing ointment came into direct contact, and thus, the hydrophobic bacterial cell wall could assist in the dissolution of hydrophobic compounds [25].

The mild retention effect of the methylcellulose hydrogel could theoretically have positively influenced the shelf-life during storage (Figure 2), but may have also ensured the release of antibacterially active compounds during application into the epidermidis (Figure 1, Table 2). In a recent study, *Cordia verbenaceae* EO-containing anti-inflammatory commercial phytotherapeutics were investigated by using the Franz cell diffusion system, which revealed that permeation profiles were strongly influenced by the carrier ointment [26]. Although, in the case of superficial skin infections, agar plate-based antibacterial testing (Figure 1) is a simple method to test efficacy, for therapeutics that target deep tissue skin infection-evoking bacteria, testing in the Franz Cell Diffusion system is an essential step in development. It seems that the aforementioned moderate retention effect of hydrogels makes them ideal candidates for EO-based applications, as was demonstrated previously with gels of *Mitracapus villosus* EO-containing cetomacrogol (BP1980), polyherbal-containing sodium carboxymethylcellulose (Na CMC), *Portulaca quadrifida*-containing carbopol 934, and hydroxy propyl methyl cellulose (HPMC) [27,28,29,30,31]. None of the above studies have analyzed the compounds released from the ointments and none of them were tested on human candidates, although these steps would be essential for further analyses. 

Overall, our data show the antibacterial properties of LEO and highlight its potential use in the treatment of PK. Further research with more human volunteers is essential to confirm these results. 

## 4. Materials and Methods

### 4.1. Bacterium Strains and Media

*Dermatophilus congolensis* (DSM 44180) and *Kytococcus sedentarius* (DSM 20547) were obtained from the German Collection of Microorganisms and Cell Cultures (DSMZ). *Bacillus thuringiensis* strain PK2021 was isolated from the PK lesions of a 43-year-old man, as described previously [14]. In addition, 10 environmental isolates of *Bacillus thuringiensis* were previously isolated by our group from environmental samples (*B. thuringiensis*_E2000/1. *B. thuringiensis*_E2001/2. *B. thuringiensis*_E2008/6. *B. thuringiensis*_E2008/7. *B. thuringiensis*_E2011/1. *B. thuringiensis*_E2016/3. *B. thuringiensis*_E2020/2. *B. thuringiensis*_E2014/1. *B. thuringiensis*_E2017/4. *B. thuringiensis*_F2020/8). All bacteria were routinely grown under aerobic conditions in Luria–Bertani Broth (LB) medium or on LB agar plates. Blood agar (BA) plates were used for *D. congolensis* (37 °C) and *K. sedentarius,* and they were grown at 37 °C and 30 °C, respectively.

### 4.2. West Indian Lemongrass Essential Oil: Gas Chromatography Analysis

West Indian lemongrass EO (LEO) (*C. citratus*) was purchased from A.G. Industries (Noida, India). To determine the compound composition, gas chromatography was performed as described previously [8]. Briefly, the Agilent 6890N/5973N GC-MSD (Santa Clara, CA, USA) system was used for the measurements using a Supelco (Sigma-Aldrich, North Brunswick, NJ, USA) SLB-5MS capillary column (30 m × 250 µm × 0.25 µm). The GC oven temperature was increased from 60 °C (3 min isothermal) to 250 °C at 8 °C/min (1 min isothermal). High-purity helium (6.0) was used as the carrier gas for the process at 1.0 mL/min (37 cm/s) in constant flow mode. The quadrupole mass analyzer was operated in full scan mode (41–500 amuat 3.2 scan/s) and in electron ionization mode at 70 eV. The MSD Chem Station D.02.00.275 software (Agilent, Santa Clara, CA, USA) was used for data evaluation. Retention data were compared with literature data and the NIST 2.0 library was also used for compound identification, while percentages were evaluated by area normalization.

### 4.3. Ointment Bases

Four different ointment bases were prepared and used for our study according to the Formulae Normales (FoNo) Editio VIII. (2021). Based on that, Hydrogelum methylcellulose FoNo VIII, Ungentum oleosum FoNo VIII. (Ung. oleoso), Unguentum stearini FoNo VIII. (Ung. stearin) and Vaselinum cholesterinatum FoNo VIII. (Vasel. cholest.) were prepared to show and compare how the different compositions and chemical properties of the base ointments influence the antibacterial efficacy of LEO. The compositions of the ointments are listed in Table 4.

One, three and five percent of LEO was added to 100 mL volumes of each ointment base, and they were used for the in vitro antibacterial testing.

### 4.4. Antibacterial Testing

To assess antimicrobial effects of EO-infused ointments, the *Bacillus thuringiensis* strain PK2021, 10 different environmental isolates of *Bacillus thuringiensis*, the *Dermatophilus congolensis* strain DSM 44180 and the *Kytococcus sedentarius* strain DSM 20547 were used. Prior to the experiments, the optical densities (ODs) of the bacteria at 600 nm (OD600) were standardized to 0.2 (~10^8^ CFU/mL) in LB medium. In each experiment, the CFUs were confirmed by performing a 100× serial dilution. Ten microliters from the dilution steps were outflowed onto the surface of LB agar medium and incubated accordingly (see Section 4.1). The next day, colonies were enumerated and multiplied back by the dilution rates. For the tests, fifty microliters from the standardized bacterial suspensions were spotted onto the surface of LB agar plates, except for *K. sedentarius* and *D. congolensis*, where BA plates were used as carriers. All bacterial suspension spots were allowed to dry. One group of them was covered with the corresponding ointment base without any LEO (positive ointment control), while group spots were covered with 50 µL volumes of different ointment bases containing 1, 3 and 5% LEO.

After 30 min of incubation, 3 × 3 mm surface areas were cut out from the treated and control spots by using a sterile spatula. These agar cubes were transferred into 300 µL PBS and vigorously vortexed for 3 × 20 s. The subsequent CFU determinations were performed by serial dilutions.

The tests were repeated three times, and the results were calculated based on the bacterial count of the control area of the 3 × 3 mm unit area and the bacterial counts recovered from the treated area after 30 min of incubation at room temperature. The antibacterial efficacies of the ointments were expressed by the rate of CFU decrease compared to that of the ointment controls containing 0% EO. Results were plotted on a graph as logarithmic values of the colony forming units.

For statistical analysis, the results of the three parallel measurements were carried out using JASP.18.3.0. Differences in response to the applied ointment bases, their EO concentrations and living bacterial numbers after tests were assessed by using a post hoc analysis with Tukey’s range test. The CFU was the dependent variable, while the fixed factors were bacteria, the EO concentration and the types of ointments. A value of *p* < 0.05 was considered significant. The *p*-value was <0.01 by the following criteria: type of ointments, bacterium CFU/mL and concentrations, as well as the type of ointments vs. bacteria, type of ointments vs. concentration, bacteria vs. concentration, and type of ointments vs. bacteria vs. concentration. The post hoc analysis was a comparison of bacteria vs. ointment type vs. concentration. We used Tukey’s test to assess the significance.

### 4.5. Shelf-Life Assessment

After preparing, the 12 different hydrogelum methylcellulose ointments (0, 1, 3, 5% LEO in three parallels) were subjected to a two-year shelf-life test. During this time, changes in the antibacterial efficacy were measured in the 6th, 12th, 18th and 24th months after preparation against the PK isolate *Bacillus thuringiensis* PK2021. For the experiment, the ointment bases were packed in 30mL ointment tubes and sealed with aluminum foil until use. The prepared tubes were stored at 23 °C until testing.

### 4.6. Franz Diffusion Cell Drug Release 

A flow-through model system was used to investigate the skin penetration of LEO emulsified in the four base ointments. The skin was represented by a synthetic membrane through which a circulating system was used to wash out the soluble components of the ointment, and samples were taken from this liquid after 30, 60, 90, 120 and 300 min. The samples were collected and the components found in them were subjected to qualitative and quantitative analysis by GC-MS.

Membrane diffusion and permeability studies were carried out with vertical Franz diffusion cell (Microette Plus, Hanson Research, Boston, MA, USA). Samples were carefully placed bubble-free into the donor compartment on the previously impregnated (pH 7.4 PBS) Sartorius^®^ polycarbonate membrane filter (Sartorius Stedim Biotech GmbH, Göttingen, Germany; pore size 0.40 μm) with an efficient diffusion surface of 1.33 cm^2^. Dissolution medium (acceptor phase) was PBS thermostated at 32 ± 0.5 °C. The mixing speed was 450 rpm. The sampling was carried out with autosampler (Hanson Microette Autosampling System, Hanson Research, USA), withdrawing 2.4 mL at every sampling time. Measurement was carried out for 5 h with 6 parallel measurements for each sample.

The samples were collected and the compound compositions of the samples obtained were analyzed by solid phase microextraction–gas chromatography–mass spectrometry (SPME-GC-MS).

### 4.7. Solid Phase Microextraction–Gas Chromatography–Mass Spectrometry

One milliliter of sample was tested in a 20-milliliter headspace (HS) bottle sealed with a silica/PTFE septum. Static vapor field analysis of the sample was performed by solid-phase micro-extraction (sHS-SPME) using a CTC Combi PAL automated sampler (CTC Analytics AG, Zwingen, Switzerland). After incubation of the sample for 5 min at 100 °C, the Stable Flex divinylbenzene/carboxene/polydimethylsiloxane (DVB/CAR/PDMS) SPME fiber (Supelco, Bellefonte, PA, USA) with a film thickness of 65 µm was introduced into the vapor space of the sample; the extraction was carried out at 100 °C for 20 min and then transferred to the injector of the gas chromatograph, where desorption took place at 250 °C for 1 min. The EO was injected in split mode (with a split ratio of 1:90), while the ointments and other liquid samples were injected in splitless mode. Finally, the fiber was cleaned and conditioned in high-purity nitrogen gas at 250 °C for 15 min.

Analysis was performed on an Agilent 6890N/5973N GC-MSD (Santa Clara, CA, USA) using a Supelco SLB-5MS capillary column (30 m × 250 µm × 0.25 µm). After an isothermal period of 3 min, the temperature of the column was raised to 60–250 °C at a rate of 8 °C/min and the final temperature was maintained for 1 min. The carrier gas was high-purity helium 6.0 and the flow rate was 1.0 mL/min (37 cm/s) in constant flow mode. Detection was performed with a quadrupole mass selective detector in electron ionization mode (70 eV), in full scan mode (41–500 amu. 3.2 scan/s). Data were evaluated using MSD Chem Station D.02.00.275 software (Agilent). For quantitative identification, retention times and mass spectra of the compounds were compared with data from standards and the NIST 2.0 library, and percentages were calculated by area normalization.

### 4.8. A Case Study to Assess the Therapeutic Efficacy of the 3% LEO-Containing Hydrogel Methylcellulose Ointment

The therapeutic efficacy of an ointment containing LEO in the treatment of PK was assessed in a single human. The patient had a long history of recurrent PK. The patient’s symptoms were misdiagnosed previously as a fungal infection, which was treated by several therapeutic courses of tetracycline and erythromycin. Prior to treatment with the 3% LEO-containing hydrogel methylcellulose ointment, the presenting bacteria were identified as previously described [14]. Based on its presence in all pits examined and its strong proteolytic activity, *B. thuringiensis* was identified as the probable etiological agent of the infection affecting both soles of the patient’s feet [14]. Treatments were applied twice a day (in the mornings and in the evenings), and before each treatment, the affected area was washed with lukewarm soapy water. After drying, the affected area was covered with a slightly thick layer of 3% LEO-containing Hydrogelum methylcellulose ointment and left to dry for at least 15 min. The treatments were repeated for 3 consecutive days. On the first, third, seventh and thirtieth day of and after the beginning of the treatment, a test evaluation of the symptoms was performed, and on the fourth day, a microbiological test was performed in order to reveal the possible presence of the *B. thuringiensis* isolate. The presence of pits and/or craters was monitored for four months after beginning the treatment. A declaration of consent was signed by the patient who was involved in the case study.

## 5. Conclusions

Based on its antibacterial spectrum on the tested etiological agents of PK, LEO has the potential to treat this superficial skin infection. The formulation is a crucial aspect, as demonstrated in vitro. Hydrogelum methylcellulose ointment base is a good candidate, as it allows active compounds to be released, but also retains them, thus ensuring a 2-year lifespan of the 3% LEO ointment. In the case of the in vivo case study, our results are promising, but further and more extensive analyses are needed with more cases. Our aim with this study was to draw attention to the antimicrobial efficacy of EO-based ointments as potential therapeutics and put forward their use as viable approved antibacterial treatments.

## Figures and Tables

**Figure 1 antibiotics-14-00241-f001:**
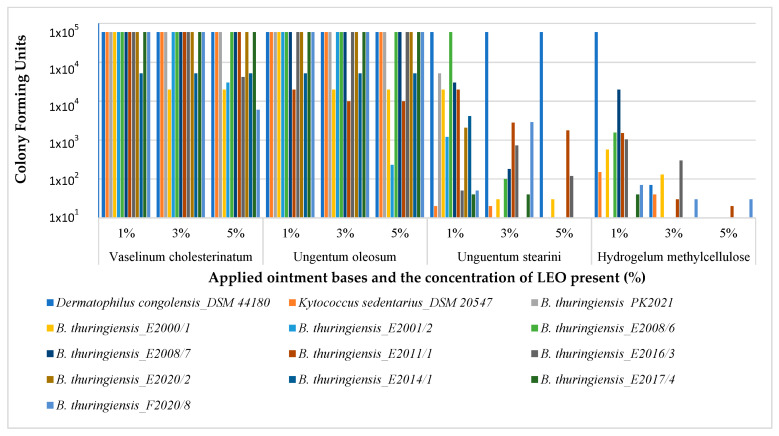
Antibacterial efficacies of increasing concentrations of LEO in different ointment bases tested on *Dermatophilus congolensis* (DSM 44180), *Kytococcus sedentarius* (DSM 20547), the *Bacillus thuringiensis strain* PK2021, and 10 natural isolates of *B. thuringiensis.* Values are expressed in colony forming units (CFUs).

**Figure 2 antibiotics-14-00241-f002:**
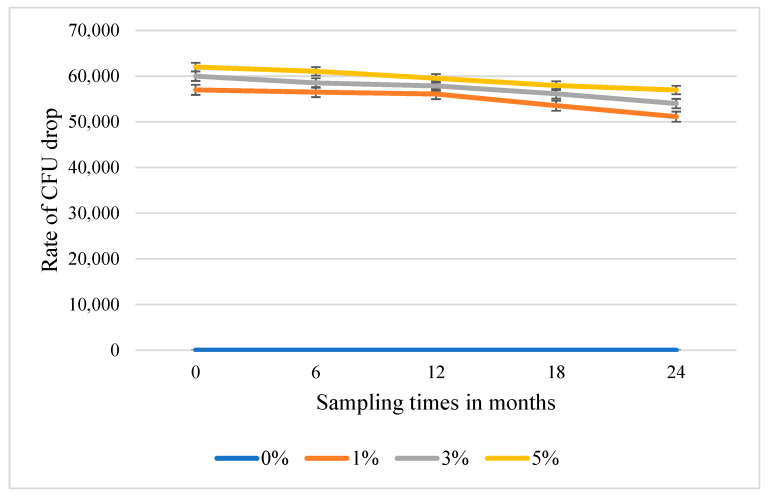
Antibacterial efficacy of Hydrogelum methylcellulose ointment containing 1%, 3%, or 5% West Indian lemongrass essential oil (*n* = 3). Antibacterial efficacy is expressed as rate of reduction in colony forming units (CFU).

**Figure 3 antibiotics-14-00241-f003:**
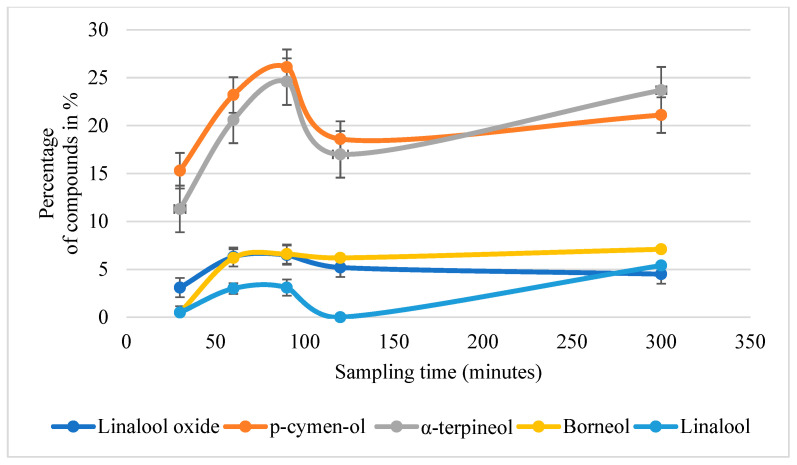
Results of the Franz diffusion cell drug release experiment. Graph shows percentage (%) recovery values of the LEO components linalool oxide, p-cymen-ol, α-terpineol, borneol and linalool released from Hydrogelum methylcellulose ointment containing 1% West Indian lemongrass essential oil at the 30th, 60th, 90th, 120th, and 300th min.

**Figure 4 antibiotics-14-00241-f004:**
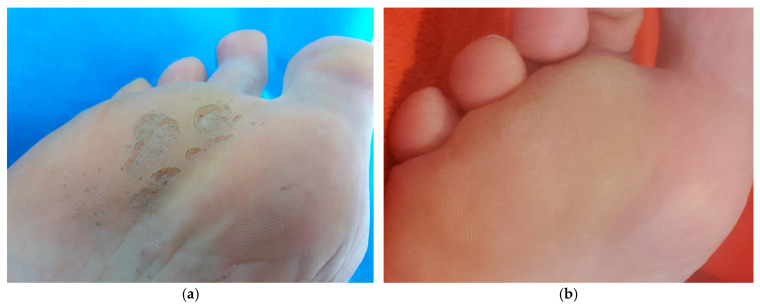
Manifestation of the visible symptoms of pitted keratolysis (**a**) before the beginning of the therapeutic treatment in the form of small pits and deep superficial crateriform erosions on the right sole and (**b**) 4 months later, after l treatment with LEO, when erosion had completely healed.

**Table 1 antibiotics-14-00241-t001:** Compound composition of the West Indian lemongrass essential oil used in the study. (Rf or retention factor stands for relative mobility of the components).

R_f_ (min)	Name	AREA %
5.1	α-Pinene	2.67
7.2	Limonene	3.28
7.3	Eucalyptol	5.48
7.4	Linalool	1.20
11.7	cis-Citral/Neral	26.45
11.9	Geraniol	2.58
12.3	trans-Citral/Geranial	27.06
12.6	Bergamotene	0.71
13.5	p-cymen-ol	4.82
14.1	α-terpineol	5.61
14.2	Borneol	5.35
14.2	Geranyl acetate	5.06
14.8	Camphene	0.96
15.1	β-Caryophyllene	2.08
15.7	Bergamotene	0.18
16.9	Piperiton	0.20
17.8	Caryophyleneoxide	3.30
		97.09

**Table 2 antibiotics-14-00241-t002:** Release time profiles of LEO compounds from four different ointment bases after incubation for 30, 60, 90, 120, and 300 mins, detected by the Franz diffusion cell drug release model. Accumulated compound release values are expressed in volume percentage and presented as means ± standard deviation (*n* = 6) (“-“: no compound was detected).

	30 min	60 min	90 min	120 min	300 min
Vaselinum cholesterinatum	-	-	-	-	-
Ungentum oleosum	-	-	-	-	β-Caryophyllene 2.0 ± 0.22
Unguentum stearini	-	-	-	-	Linalool 7.1 ± 0.56, deriv of linalool 4.0 ± 1.56, Camphene 0.6 ± 0.50, β-Caryophyllene 6.7 ± 1.87
Hydrogelum methylcellulose	Linalool-oxide 3.1 ± 1.24, p-cymen-ol 15.3 ± 2.39, α-terpineol 11.3 ± 3.14	Linalool-oxide 6.3 ± 1.32, p-cymen-ol 23.2 ± 1.28, α-terpineol 20.6 ± 1.18, Borneol 6.2 ± 0.65, Linalool 3.0 ± 0.31, Piperiton 2.1 ± 0.77	Linalool-oxide 6.5 ± 1.84, p-cymen-8-ol 26.1 ± 0.57, α-terpineol 24.6 ± 1.61, Borneol 6.6 ± 0.90, Linalool 3.1 ± 0.57, Camphene 1.0 ± 0.25,	Linalool-oxide 5.2 ± 1.52, p-cymen-ol 18.6 ± 1.0, α-terpineol 17.0 ± 4.26, Borneol 6.2 ± 1.00,	Linalool-oxide 4.5 ± 1.25, p-cymen-ol 21.1 ± 2.61, α-terpineol 23.7 ± 2.91, Borneol 7.1 ± 0.24, Linalool 5.4 ±0.85, Piperiton 2.1 ± 0.28

**Table 3 antibiotics-14-00241-t003:** Experimental therapeutic efficacy of Hydrogel methylcellulose ointment containing 3% LEO. Ratings qualitatively rank symptoms of PK before and throughout treatment. Microbiological testing for the presence of the etiological agent *B. thuringiensis* was performed only on the fourth day. (Sign and test explanation: Scale and intensity ratings of malodor: “+++” a penetrant smell from the foot after removing the shoe, “++” odor detected from foot after removing the socks, “-” no smell at all. Sensitivity: “+++” unpleasant sensitivity at rest, “++” unpleasant sensitivity under stationary load, “+” unpleasant sensitivity under rhythmic loads (running, walking), “-” no sensitivity. Itching: “++” itching under stationary load, “+” itching under rhythmic loads (running, walking), “-” no itching. Burning sensation: burning feeling at rest, “++” burning feeling under stationary load, “+” burning feeling under rhythmic loads (running, walking), “-” no burning feeling. n.a.: not adequate; n.p.: not performed.)

	Number of Days Since the Start of the Treatment						
Symptoms	0	1	3	4	7	30	120
Malodor	+++	++	-	n.p.	-	-	-
Sensitivity	+++	++	+	n.p.	-	-	-
Itching	++	+	-	n.p.	-	-	-
Burning sensation	++	+	-	n.p.	-	-	-
Presence of pits	+	+	+	+	+	+	-
Novel pit formation	n.a.	-	-	n.p.	-	-	-
Microbiological testing	n.p.	n.p.	n.p.	-	n.p.	n.p.	n.p.

**Table 4 antibiotics-14-00241-t004:** Composition and general characteristics of the ointment bases used in this study. LEO was added to the ointment bases at the rates of 1, 3, and 5%.

Vaselinum Cholesterinatum	Ungentum Oleosum	Unguentum Stearini	Hydrogelum Methylcellulose
Alcohol cetylicus 30 g Cera alba 30 g Adepslanae 50 g Vaselinum album 870 g Cholesterolum 20 g	Alcoholesadipislanae 50 g Cera alba 100 g Alcohol cetylicus et stearylicus 150 g Ricinioleum virginale 700 g	Alcohol cetylicus et stearylicus 45 g Acidum stearicum 100 g Natrii laurilsulfas 5 g Solutio conservans 10 g Sorbitolum 35 g Gylcerolum 85% 100 g Aqua purificata 705 g	Methylcellulosum 50 g Gylcerolum 85% 100 g Solutio conservans 10 g Aqua purificata 840 g
Mixture of lipophilic ingredients and cholesterol as an emulsifier.	Mostly lipophilic ingredients. containing emulsifier and minimally water-soluble.	Oil/water emulsion that is relatively water-soluble.	Hydrophilic, water-soluble.

## Data Availability

All data are presented in the manuscript.

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
