# Peer review of "The Therapeutic Potential of West Indian Lemongrass (Cymbopogon citratus) Essential Oil-Based Ointment in the Treatment of Pitted Keratolysis"

_antibiotics, 2025, doi:10.3390/antibiotics14030241_

Round 1
Reviewer 1 Report
Comments and Suggestions for Authors
The manuscript titled “Therapeutic Potential of West Indian Lemongrass (Cymbopogon citratus) Essential Oil-Based Ointment in the Treatment of Pitted Keratolysis” presents an innovative idea, exploring the potential use of lemongrass essential oil (LEO) as an alternative to antibiotics for treating pitted keratolysis. While the concept is promising and highly relevant, the manuscript is hindered by significant methodological and scientific shortcomings that impact the reliability and validity of its findings. As it stands, the manuscript does not meet the publication standards of Antibiotics.
Several concerns:
1. The in vivo study relies on a single patient, with no inclusion of a control group or randomization. This significantly limits the reliability and generalizability of the findings regarding the ointment’s efficacy, as responses to treatments vary significantly between individuals.
2. Key experiments, such as antibacterial efficacy and Franz diffusion tests, lack statistical analyses, leaving the strength and significance of the results uncertain.
3. The manuscript does not mention obtaining ethical approval or informed consent, which are essential for studies involving human participants. Is EO used have passed the toxicity regulation?
4. The manuscript claims that lemongrass essential oil can replace antibiotics, which is an overgeneralization given the limited scope of the study and lack of comparative data.
5. The conclusions are not sufficiently supported by the data provided, especially given only 1 participant in vivo experiment.
6. Figures and tables lack clarity and detail (e.g., missing error bars, small fonts, and unclear thresholds for significant results).
7. The GC-MS results are presented, but their relevance to antibacterial activity is not well-explained or linked to the overall findings.
Comments on the Quality of English LanguageProofread by professional English native speaker is needed to refine the manuscript.
Author Response
reviewer 1.
Detailed Responses to Editor and Reviewers
Antibiotics ID - 3416675
Response to the Reviewer’s comments
We are very pleased to resubmit for publication the revised version of “Therapeutic potential of West Indian lemongrass (Cymbopogon citratus) essential oil-based ointment in the treatment of pitted keratolysis.” by Schneider Gy, Schweitzer B, Steinbach A, Solti-Hodován Á, Horváth M, Bakó Eszter, Mayer A, and Pál Szilárd for consideration for publication in MDPI-Antibiotics in to special issue “Antimicrobial and Anti-infective Activity of Natural Products, 2nd Edition”. We are grateful for your positive opinion and about our manuscript. We considered your comments hoping our revision has improved the paper to a level of your satisfaction.
Here we present your comments (R1), while we give our response in (A) in red bold.
Response to Reviewer #1 Comments and suggestions
Comments and Suggestions for Authors
The manuscript titled “Therapeutic Potential of West Indian Lemongrass (Cymbopogon citratus) Essential Oil-Based Ointment in the Treatment of Pitted Keratolysis” presents an innovative idea, exploring the potential use of lemongrass essential oil (LEO) as an alternative to antibiotics for treating pitted keratolysis. While the concept is promising and highly relevant, the manuscript is hindered by significant methodological and scientific shortcomings that impact the reliability and validity of its findings. As it stands, the manuscript does not meet the publication standards of Antibiotics.
A: Dear reviewer thank you for your time to read our manuscript and we try to repsond for your the critics and remarks.
Several concerns:
- The in vivo study relies on a single patient, with no inclusion of a control group or randomization. This significantly limits the reliability and generalizability of the findings regarding the ointment’s efficacy, as responses to treatments vary significantly between individuals.
A: At this case the causative agent was determined and then the treatment was performed. Yes you are right and therefore our experiments are in the category of a case file. We certainly do not want to overstate the result of this in vivo test, but shows that the LEO based ointment might be effective not only in vitro but also in vivo. Certainly more tests/individuals are required in the future to get strongly supported results. Based on our results this we perform with the Dermatology Department at our university.
- Key experiments, such as antibacterial efficacy and Franz diffusion tests, lack statistical analyses, leaving the strength and significance of the results uncertain.
A: The antibacterial tests were performed in three parallels and results were present in a graph (Figure 1). Now we performed the statistic analysis that we present at the relevant sectipons (MM, R, Supplementary data1). For that we performed Post-hoc statistical analysis with the Tukey’s range test.
In case of the Franz diffusion drug release model the statistic analyses was also performed based on the 6 parallel data, and according to that the data in the relevant table were expanded with the deviations.
- The manuscript does not mention obtaining ethical approval or informed consent, which are essential for studies involving human participants. Is EO used have passed the toxicity regulation?
A: After submission this declaration was required that we than sent to the Editors offic. In that certainly the patient agreed that the experimental ointment would be used to treat out his PK infection and results can be published, certainly without mentioning the identity of te patient.
- The manuscript claims that lemongrass essential oil can replace antibiotics, which is an overgeneralization given the limited scope of the study and lack of comparative data.
A: In our manuscript we wrote „in certain case”. In abstract we strictly focused on PK as we wrote that „…West Indian lemongrass essential oil (LEO) is a potential alternative to replace antibiotics in the treatment of pitted keratolysis.”. We did not want to generalise as we are aware that still for a long time antibiotics will be the drugs of choice in most of the cases, but if we want decrease the usage of antibiotics and by this we want to reduce the environmental presence of antibiotics and by that reduce the development of antibiotic resistance, than EOs should be considered as alternatives - in certain cases. PK can be one target for this.
- The conclusions are not sufficiently supported by the data provided, especially given only 1 participant in vivo experiment.
A: In accordance with your remark, we have been more cautious in our wording regarding the in vivo result.
- Figures and tables lack clarity and detail (e.g., missing error bars, small fonts, and unclear thresholds for significant results).
A: In case of Figure 2 error bars are not presented becasue of the sizes of the columns. If we put error bars they will disturb the clarity. Also in case of this figure we increased font and letter sizes. We did the same with Figure 2 and Table 2..
- The GC-MS results are presented, but their relevance to antibacterial activity is not well-explained or linked to the overall findings.
A: GC analysis is a kind of quality control that is necessary to perform as the compound composition of different EO batches can be slightly different. We added this information now into Discussion.
Comments on the Quality of English Language
Proofread by professional English native speaker is needed to refine the manuscript.
Submission Date
23 December 2024
Date of this review
31 Dec 2024 15:13:46
Reviewer 2 Report
Comments and Suggestions for Authors
The article is of interest for treating a pathology (PK) with a conventional treatment of antibiotics. The research proposes formulations with lemongrass essential oil as an alternative that is justified acceptably. The selection of the formulations from Formulae Normales is correct since it allows us to compare the inclusion and release of LEO from the product. The design of the in vitro antimicrobial tests is appropriate since it uses microorganisms linked to PK pathology. The monitoring of the antimicrobial activity over time (half-life test) is convincing. The study of the chemical composition of the LEO using a chromatographic profile by GC-MS provides sufficient information. The in vitro release test with Franz cells allows us to compare the release of components from the formulations and the method used to recognize the released components is valid. The in vivo study with a patient should be reconsidered since it is insufficient for determining therapeutic efficacy. In the attached file I propose some suggestions for revision in the manuscript.
Article: Therapeutic potential of West Indian lemongrass (Cymbopogon 2 citratus) essential oil-
based ointment in the treatment of pitted keratolysis.
Authors: Schneider G, Schweitzer B, Steinbach A, Solti-Hodován Á, Horváth M, Bakó Eszter,
Mayer A, and Pál Szilárd.
Comments
In Abstract, the objective of the work is not expressed. There, the results of the shelf life of the
ointments are not mentioned and the conclusions are not adequately stated.
In the introduction, the first time indicate the full scientific name of the plant: Genus species,
abbreviation of the Classifier (Family).
In Table 1, formulation of Vaselinum cholesterinatum, see “Mixture of lipophilic ingredients
that contain cholesterinas an emulsifier and therefore watersoluble”, look at “colesterinas”
would have to change for “cholesterol as emulsifier”. The formulation is a classic example of an
anhydrous absorption base, which allows the incorporation of water or aqueous solutions to
form water-in-oil (W/O) emulsions. The expression “soluble in water” is confusing, it leads to
thinking of easily soluble in water, and the absorption bases are not easily removed from the
skin with water washing. The description could say: “the mixture contains lipophilic ingredients
and cholesterol as an emulsifier, resulting in a hydrophilic base”. This does not mean it is easily
washable with water because the external phase of the emulsion is oleaginous.
In Materials and Methods, item 4.3 - lines 294-295, It is not clear: “The antibacterial efficacy of
the ointments was expressed as multiplicative values and plotted on a graph”. In this case,
effectiveness was assessed by the reduction in bacterial numbers, expressed as log colony-
forming units, CFU/cm3 or CFU/ml. Therefore, in materials and methods, the mode of
expressing the results of the antimicrobial activity should be defined more clearly.
In Materials and Methods, the following is indicated: “180 μl of Britton-Robinson buffer
solution containing 1 v/v% DMSO stock solution at different pH were measured into the cells of
the bottom plate of the PAMPA sandwiches”. The results do not describe any observations or
data regarding the effect of different pHs on the release of active components. It is necessary
to determine whether this comparison at different pHs is relevant.
In Materials and Methods, lines 365-366, “the therapeutic efficacy of an ointment based on
lemongrass essential oil in the treatment of pitted keratolysis was tested in a patient with a
long history of recurrent PK”. The in vivo study of a patient is insufficient to obtain evidence of
the therapeutic efficacy of the formulation. Efficacy is typically measured using statistical
methods in the context of a clinical trial. Also, the goal is always to determine whether the
treatment has a significant effect compared to a control group. This was not raised in the study
described. This test could be justified as a case study, where the patient presents a chronic
pathology and the evolution of symptoms with the treatment of the ointment under study is
determined.
In Materials and Methods, lines 376-377, indicated “The treatments were repeated for 3
consecutive days and microbiological examinations of the pits were performed”.
The results do not describe observations or values of antimicrobial activity, instead the table 3
details the evolution of symptoms in 30 days. This should be reviewed and the description of
the method should be in accordance with the results presented.
In Results, line 170, the figure 3-b) shows an image of the evolution of the lesions at 4 months.
In materials and methods do not indicate that the evolution of the pathology is followed over
time. Please review this and clarify in methods.
The conclusions are presented in a general way and it is convenient to recall that in the
comparison of formulations, one resulted with antimicrobial efficacy in the in vitro tests, that a
concentration with a better result of antimicrobial activity could be determined and its useful
life in 2 years was acceptable.
Author Response
Reviewer 2.
Detailed Responses to Editor and Reviewers
Antibiotics ID - 3416675
Response to the Reviewer’s comments
We are very pleased to resubmit for publication the revised version of “Therapeutic potential of West Indian lemongrass (Cymbopogon citratus) essential oil-based ointment in the treatment of pitted keratolysis.” by Schneider Gy, Schweitzer B, Steinbach A, Solti-Hodován Á, Horváth M, Bakó Eszter, Mayer A, and Pál Szilárd for consideration for publication in MDPI-Antibiotics in to special issue “Antimicrobial and Anti-infective Activity of Natural Products, 2nd Edition”. We are grateful for your positive opinion and about our manuscript. We considered your comments hoping our revision has improved the paper to a level of your satisfaction.
Here we present your comments (R2), while we give our response in (A) in red bold.
Response to Reviewer #2 Comments and suggestions
R2:
Comments and Suggestions for Authors
The article is of interest for treating a pathology (PK) with a conventional treatment of antibiotics. The research proposes formulations with lemongrass essential oil as an alternative that is justified acceptably. The selection of the formulations from Formulae Normales is correct since it allows us to compare the inclusion and release of LEO from the product. The design of the in vitro antimicrobial tests is appropriate since it uses microorganisms linked to PK pathology. The monitoring of the antimicrobial activity over time (half-life test) is convincing. The study of the chemical composition of the LEO using a chromatographic profile by GC-MS provides sufficient information. The in vitro release test with Franz cells allows us to compare the release of components from the formulations and the method used to recognize the released components is valid. The in vivo study with a patient should be reconsidered since it is insufficient for determining therapeutic efficacy. In the attached file I propose some suggestions for revision in the manuscript.
A: First of all thank You very much for your positive opinion and comments. Here we try to respond for your critics and questions.
R2.: Comments
- In Abstract, the objective of the work is not expressed. There, the results of the shelf life of the
ointments are not mentioned and the conclusions are not adequately stated.
A: We reformulated the Abstract in a more direct way, and also flashed up the results of the shelf life experiments. Beside we also reformulated conclusion.
- In the introduction, the first time indicate the full scientific name of the plant: Genus species, abbreviation of the Classifier (Family).
A: In abstract at the first place we left the complete scientific name and later we used the shortened version (C. citratus)
- In Table 1, formulation of Vaselinum cholesterinatum, see “Mixture of lipophilic ingredients
that contain cholesterinas an emulsifier and therefore watersoluble”, look at “colesterinas”
would have to change for “cholesterol as emulsifier”. The formulation is a classic example of an
anhydrous absorption base, which allows the incorporation of water or aqueous solutions to
form water-in-oil (W/O) emulsions. The expression “soluble in water” is confusing, it leads to
thinking of easily soluble in water, and the absorption bases are not easily removed from the
skin with water washing. The description could say: “the mixture contains lipophilic ingredients
and cholesterol as an emulsifier, resulting in a hydrophilic base”. This does not mean it is easily
washable with water because the external phase of the emulsion is oleaginous.
A: Thank You for the precise calirification! We corrected according to your suggestions. So it is clear.
- In Materials and Methods, item 4.3 - lines 294-295, It is not clear: “The antibacterial efficacy of the ointments was expressed as multiplicative values and plotted on a graph”. In this case,
effectiveness was assessed by the reduction in bacterial numbers, expressed as log colonyforming units, CFU/cm3 or CFU/ml. Therefore, in materials and methods, the mode of expressing the results of the antimicrobial activity should be defined more clearly.
A: Corrections were made in the text (now it is in 4.4, L333-336) and additions were given in the relevant Figure 1.
- In Materials and Methods, the following is indicated: “180 μl of Britton-Robinson buffer
solution containing 1 v/v% DMSO stock solution at different pH were measured into the cells of
the bottom plate of the PAMPA sandwiches”. The results do not describe any observations or
data regarding the effect of different pHs on the release of active components. It is necessary
to determine whether this comparison at different pHs is relevant.
A: No it is not relevant. In the Franz Diffusion cell drug release assay experiments were only performed only on pH 7.4. The text is corrected now.
- In Materials and Methods, lines 365-366, “the therapeutic efficacy of an ointment based on
lemongrass essential oil in the treatment of pitted keratolysis was tested in a patient with a
long history of recurrent PK”. The in vivo study of a patient is insufficient to obtain evidence of
the therapeutic efficacy of the formulation. Efficacy is typically measured using statistical
methods in the context of a clinical trial. Also, the goal is always to determine whether the
treatment has a significant effect compared to a control group. This was not raised in the study
described. This test could be justified as a case study, where the patient presents a chronic
pathology and the evolution of symptoms with the treatment of the ointment under study is
determined.
A: Dear reviewer, yes we are aware of the shortcomings of the test as it was a single case, with that person from whom the Bacillus thuringiensis strain was isolated. Please have look on it, as we reformulated the text (4.8), if it is satisfactory for emphasizing that this was only a single test, so actually a case test. Certainly we are in contact with the Dermatology Department and in case of new cases we will test the oinment further.
- In Materials and Methods, lines 376-377, indicated “The treatments were repeated for 3
consecutive days and microbiological examinations of the pits were performed”.
The results do not describe observations or values of antimicrobial activity, instead the table 3
details the evolution of symptoms in 30 days. This should be reviewed and the description of
the method should be in accordance with the results presented.
A: We synchronised MM (4.8), and data in Table 3., by considering your remark.
- In Results, line 170, the figure 3-b) shows an image of the evolution of the lesions at 4 months.
In materials and methods do not indicate that the evolution of the pathology is followed over
time. Please review this and clarify in methods.
A: Yes it was a little bit disturbing. Now we synchronised MM and Table 3. and Figure 3.
The conclusions are presented in a general way and it is convenient to recall that in the
comparison of formulations, one resulted with antimicrobial efficacy in the in vitro tests, that a
concentration with a better result of antimicrobial activity could be determined and its useful
life in 2 years was acceptable.
A: We reformulated Conclusions and now the aspects You mentioned are also involved.
Thank You for your thorough help and suggestions!
Submission Date
23 December 2024
Date of this review
09 Jan 2025 04:36:27
Reviewer 3 Report
Comments and Suggestions for Authors
This manuscript presents applied research-oriented experiments for the development of a lemongrass essential oil containing ointment can be used against the symptoms of pitted keratolysis. With this context, antimicrobial activity of lemongrass essential oil in different base ointments has been evaluated, as well as the solubility of different essential oil compounds in ointment bases has been studied. Presentation and evaluation of the results are acceptable; however, I have some comments for the Authors to further polish the manuscript:
- Italicize the genus and species names. Please check the manuscript with this regard.
- The Introduction part should be supplemented with some informations on other natural approaches previously tested against pitted keratolysis.
- Figures 1 and 2: Presentation of the results in Log CFU data would be better in my opinion.
- Figure 1: Re-edit the column graph in order to see the top of columns that reached the figure limit. But I believe this step won’t be necessary if the data is presented as Log CFU.
- There is no in-text citation for Figure 2.
- Table 3: Define as footnote what is the scale/intensity that the signs "+++", "++" etc. represent.
- No in-text citation to Figure 3 is described in the section Results. Please correct.
- L199-200: How can these carriers work in case of other compounds, antimicrobial agents, essential oils, etc.? Can these carriers be applicable for other molecules? I miss a brief comparison with other tests in discussion part.
- L230: Are there other practical applications for hydrogel containing essential oil? A brief discussion should be inserted for this issue as well.
- L263: Confusing subchapter title, please clarify. The Authors should describe each assay in individual subchapters.
- L273: This is Table 4. Please correct.
- L291: How was the CFU determination performed? There is no description of this methodology.
- L330: Unlock the abbreviation „NKVR”.
- The section Conclusion contains some general sentences. It does not present exact conclusions from the results. So, please supplement this section with relevant conclusions drawn from the results of the study.
Author Response
Reviewer 3.
Detailed Responses to Editor and Reviewers
Antibiotics ID - 3416675
Response to the Reviewer’s comments
We are very pleased to resubmit for publication the revised version of “Therapeutic potential of West Indian lemongrass (Cymbopogon citratus) essential oil-based ointment in the treatment of pitted keratolysis.” by Schneider Gy, Schweitzer B, Steinbach A, Solti-Hodován Á, Horváth M, Bakó Eszter, Mayer A, and Pál Szilárd for consideration for publication in MDPI-Antibiotics in to special issue “Antimicrobial and Anti-infective Activity of Natural Products, 2nd Edition”. We are grateful for your positive opinion and about our manuscript. We considered your comments hoping our revision has improved the paper to a level of your satisfaction.
Here we present your comments (R2), while we give our response in (A) in red bold.
Response to Reviewer #3 Comments and suggestions
R3.:
Comments and Suggestions for Authors
This manuscript presents applied research-oriented experiments for the development of a lemongrass essential oil containing ointment can be used against the symptoms of pitted keratolysis. With this context, antimicrobial activity of lemongrass essential oil in different base ointments has been evaluated, as well as the solubility of different essential oil compounds in ointment bases has been studied. Presentation and evaluation of the results are acceptable; however, I have some comments for the Authors to further polish the manuscript:
A.: Thank You for your positive opinion and help!
- Italicize the genus and species names. Please check the manuscript with this regard.
A.: We made it.
- The Introduction part should be supplemented with some informations on other natural approaches previously tested against pitted keratolysis.
A.: we certainly searched for such articles by using the words “pitted keratolysis”, natural, essential oil, remedy”, in different combinations, but did not find any that was based on natural remedies, only on traditional antibiotics.
- Figures 1 and 2: Presentation of the results in Log CFU data would be better in my opinion.
A.: In case of Figure 1. we changed the scale to logarithmic so You will find in the text now, but in case of Figure 2 (shelf life) we find the original graph more spectacular as some tendencies (slight loss of efficacies) can be suggested better. Therefore we left the old version in the text in this latter case. But for seeing the difference we present here the logarithmic graph version for the shelf life (Figure 2).
- Figure 1: Re-edit the column graph in order to see the top of columns that reached the figure limit. But I believe this step won’t be necessary if the data is presented as Log CFU.
A.: we made it according to your suggestion.
- There is no in-text citation for Figure 2.
A.: Thank You we inserted it now. (L268)
- Table 3: Define as footnote what is the scale/intensity that the signs "+++", "++" etc. represent.
A.: explanation for the sensations are added.
- No in-text citation to Figure 3 is described in the section Results. Please correct.
A.: It is added now. Thank You! (L176)
- L199-200: How can these carriers work in case of other compounds, antimicrobial agents, essential oils, etc.? Can these carriers be applicable for other molecules? I miss a brief comparison with other tests in discussion part.
A.: Actually we did not find articels with these ointment bases, that are basic ointments in dermatological practice but actually listed some examples. (L230-234)
- L230: Are there other practical applications for hydrogel containing essential oil? A brief discussion should be inserted for this issue as well.
A.: We have found some publications with other gel formats (carbopol etc.), that we inserted into the discussion part. (L276-281)
- L263: Confusing subchapter title, please clarify. The Authors should describe each assay in individual subchapters.
A.: This MM part we divided in three different subchapters based on your suggestion.
- L273: This is Table 4. Please correct.
A.: Corrected, thank You!
- L291: How was the CFU determination performed? There is no description of this methodology.
A.: We presented now in the relevant section you mentioned (L333-336)
- L330: Unlock the abbreviation „NKVR”.
A.: NKVR is Hungarian term and stayed there accidently, that is actually not relevant to the Franz diffusion assay. Thank You for finding it! We deleted.
- The section “Conclusion” contains some general sentences. It does not present exact conclusions from the results. So, please supplement this section with relevant conclusions drawn from the results of the study.
A.: We reformulated this section. I think now the major issues are emphasized better.

Reviewer 4 Report
Comments and Suggestions for Authors
The authors submitted a research article of interest that evaluated the therapeutic potential of Cymonbogon (lemon grass) essential oil ointments in dermatology. The authors explored, with particular emphasis, the antimicrobial activity of such ointments. The subject of the present manuscript is of particular pertinence within the pertinent field, given that the utilization of antimicrobial ointments has the potential to curtail the use of antibiotics in the domain of dermatology.
Given the current widespread use of antibiotics, which has led to a proliferation of antibiotic-resistant bacterial strains, the development of alternative antibacterial treatment approaches is imperative and urgent. The authors of the present study sought to ascertain whether the utilization of essential oils could supplant antibiotic treatments in any aspect of infection therapy and impede the proliferation of antibiotic-resistant microbial species.
The authors employed an adequate and well-established set of methods. Consequently, the methodology proved to be effective. A meticulous investigation was conducted into the effects of various ointment compositions containing Cymonbogon essential oil on bacterial growth. The findings are presented in figures and tables to enhance the clarity of the text. Furthermore, the outcomes substantiate the conclusions. Furthermore, the authors have juxtaposed their theories, arguments, and discussions with relevant references. As a non-native speaker of English, I am unable to assess any linguistic aspects of the manuscript text.
Author Response
Reviewer 4
Detailed Responses to Editor and Reviewers
Antibiotics ID - 3416675
Response to the Reviewer’s comments
We are very pleased to resubmit for publication the revised version of “Therapeutic potential of West Indian lemongrass (Cymbopogon citratus) essential oil-based ointment in the treatment of pitted keratolysis.” by Schneider Gy, Schweitzer B, Steinbach A, Solti-Hodován Á, Horváth M, Bakó Eszter, Mayer A, and Pál Szilárd for consideration for publication in MDPI-Antibiotics in to special issue “Antimicrobial and Anti-infective Activity of Natural Products, 2nd Edition”. We are grateful for your positive opinion and about our manuscript. We considered your comments hoping our revision has improved the paper to a level of your satisfaction.
Here we present your comments (R2), while we give our response in (A) in red bold.
Response to Reviewer #1 Comments
Opinion of Reviewer 1.:
The authors submitted a research article of interest that evaluated the therapeutic potential of Cymonbogon (lemon grass) essential oil ointments in dermatology. The authors explored, with particular emphasis, the antimicrobial activity of such ointments. The subject of the present manuscript is of particular pertinence within the pertinent field, given that the utilization of antimicrobial ointments has the potential to curtail the use of antibiotics in the domain of dermatology.
Given the current widespread use of antibiotics, which has led to a proliferation of antibiotic-resistant bacterial strains, the development of alternative antibacterial treatment approaches is imperative and urgent. The authors of the present study sought to ascertain whether the utilization of essential oils could supplant antibiotic treatments in any aspect of infection therapy and impede the proliferation of antibiotic-resistant microbial species.
The authors employed an adequate and well-established set of methods. Consequently, the methodology proved to be effective. A meticulous investigation was conducted into the effects of various ointment compositions containing Cymonbogon essential oil on bacterial growth. The findings are presented in figures and tables to enhance the clarity of the text. Furthermore, the outcomes substantiate the conclusions. Furthermore, the authors have juxtaposed their theories, arguments, and discussions with relevant references. As a non-native speaker of English, I am unable to assess any linguistic aspects of the manuscript text.
Answer: Thank You for your supportive opinion!
Round 2
Reviewer 1 Report
Comments and Suggestions for Authors
The revisions do not sufficiently address the concerns raised in the previous review. The changes made are minimal and do not significantly improve the manuscript in terms of clarity, depth, or scientific contribution.
Author Response
Dear reviewer 1.
Concerning to your critics we provide our answers bellow:
Comments 1
The in vivo study relies on a single patient, with no inclusion of a control group or randomization. This significantly limits the reliability and generalizability of the findings regarding the ointment’s efficacy, as responses to treatments vary significantly between individuals.
Response 1.
Pitted keratolysis (PK) is a typical skin surface infection, caused by bacteria. The atiological agents only reside on the stratum corneum and do not invade deep tissues. By that a therapeutic agent of PK has to be effective on the surface, like in the case of the currently used antibiotics erythromycin and clindamycin. It means that the agar based inhibition test is a proper model for efficacy testing, as during treatment not the person himself is treated as in the case of an internal medicine, but the bacterium that colonises the stratum corneum. The most important question here is to demonstrate if the ointment has the capacity to clear the bacteria from a surface, and it was effective.
Concerning to the in vivo test we had the possibility to test the LEO containing ointment. This was certainly only one case, similarly to other case studies published in other journals like JAAD Case Reports or in European Journal of Dermatology. So this is not unusual aspect of research, as these case studies in several cases do not use control groups (due to the limitation or rarity of the cases forexample), but represent an individual case and draws attention that a certain treatment could be effective and worth to study further.
Comments 2.
Key experiments, such as antibacterial efficacy and Franz diffusion tests, lack statistical analyses, leaving the strength and significance of the results uncertain.
Response 2.
For the antibacterial test we provided the relevant statistical analysis (Figure1), according to that three parallel measurements were carried out in order to test the antibacterial efficacies of the different ointment bases, containing 1%, 3% and 5% LEO concentrations. For the statistical analysis, results of these 3 experiments were used. A value of p < 0.05 was considered significant. During the analysis, the CFU was the dependent variable, while the fixed factors were bacteria, the concentration and the types of the ointments. P-value was <0.01 by the following criteria: type of ointments, bacterium CFU and concentrations, moreover by the type of oitments*bacteria, type of ointments*contentration, bacteria*concentration, and type of ointments*bacteria*contentration. The post-hoc analysis was a comparison of bacteria*ointment type*concentration. We used Tukey's test to assess the significance.
This detailed method is described now in MM 4.4 and in Supplementary Data 1.
In frame of the Franz diffusion drug release test 6 parallel measurements were carried out and based on the gained percentage values data were presented in Table 2. Standard deviations of the released quantities at the relavant sampling times were determined and were indicated beside the data of table 1. Furthermore gained release characteristics are also visualized now in Fugure 3.
Comments 3.
The manuscript does not mention obtaining ethical approval or informed consent, which are essential for studies involving human participants. Is EO used have passed the toxicity regulation?
Response 3.
This issue is mentioned now in the manuscript in the last sentence of MM 4.8. The consent was submitted earlier in to the editorial office.
Comments 4.
The manuscript claims that lemongrass essential oil can replace antibiotics, which is an overgeneralization given the limited scope of the study and lack of comparative data.
Response 4.
You are absolutely right, we did not want to use this sentence in a generalised context, but strictly on the treatment of skin surface infections. Based on this we completed the sentence.
Comments 5.
The conclusions are not sufficiently supported by the data provided, especially given only 1 participant in vivo experiment.
Response 5.
We reformulated the Conclusion part in the previous review process and there we stated two important issues according to your remark:
- Based on its antibacterial spectrum on the tested aetiological agents of pitted keratolysis the West Indian lemongrass essential oil has a potential to treat this superficial skin infection.
- In case of the in vivo case study our results are promising, but further and more extensive analyses are needed, with more cases.
Comments 6.
Figures and tables lack clarity and detail (e.g., missing error bars, small fonts, and unclear thresholds for significant results).
Response 6.
We changed font sizes, showed error bars and presented the required statistical analysis (please see our answers to your second review point).

Reviewer 3 Report
Comments and Suggestions for Authors
The Authors have improved the manuscript. It can be accepted after clarifying some points concerning the data presentation on Figure 1 and statistical analysis.
- It is unclear what statistical analysis results are depicted in Figure 1B. Please explain more precisely what Figure 1B represents.
- How was the statistical analysis performed? How many parallel studies have been performed? What P value was considered statistically significant? In the supplementary file: on what basis does the author write that some data are statistically different or not? Present more data and methodology detail concerning the statistical analysis.
- Also, in Figure 1, the label for the y-axis is still incomprehensible to me. What is the reason indicating the „rate of CFU drop” instead of presentation of exact cell numbers in Log CFU?
Author Response
The Authors have improved the manuscript. It can be accepted after clarifying some points concerning the data presentation on Figure 1 and statistical analysis.
Comment 1
It is unclear what statistical analysis results are depicted in Figure 1B. Please explain more precisely what Figure 1B represents.
Response 1
This figure showed how the different ointment bases generally affected the CFU decreases of the tested bacteria. The descriptive plot was performed with the JASP statistical program. On the horizontal axis of the graph’s the applied ointments are shown, while the lines represent the rate of CFU drop values. It showed that the highest rate of CFU decrease was generated by the 5% Hydrogelum methylcellulose and further on. Since based on your suggestion we replaced Figure 1., we also put the recently generated Figure 1B out from the manuscript.
Comment 2
How was the statistical analysis performed? How many parallel studies have been performed? What P value was considered statistically significant? In the supplementary file: on what basis does the author write that some data are statistically different or not? based on the p values, as Present more data and methodology detail concerning the statistical analysis. The following text was added to Suppl Data 1.
Response 2
Three parallel measurements were carried out in order to test the antibacterial efficacies of the different ointment bases, containing 1%, 3% and 5% LEO concentrations. For the statistical analysis, results of these 3 experiments were used. A value of p < 0.05 was considered significant. During the analysis, the CFU was the dependent variable, while the fixed factors were bacteria, the concentration and the types of the ointments. P-value was <0.01 by the following criteria: type of ointments, bacterium CFU and concentrations, moreover by the type of oitments*bacteria, type of ointments*contentration, bacteria*concentration, and type of ointments*bacteria*contentration. The post-hoc analysis was a comparison of bacteria*ointment type*concentration. We used Tukey's test to assess the significance.
This detailed method is described now in MM 4.4.
Comment 3
Also, in Figure 1, the label for the y-axis is still incomprehensible to me. What is the reason indicating the „rate of CFU drop” instead of presentation of exact cell numbers in Log CFU?
Response 3
According to your suggestion, we changed this figure and in this version the direct CFU numbers are presented in the graph.
Dr. György Schneider
corresponding author
